# A Study of Gentianae Radix et Rhizoma Class Differences Based on Chemical Composition and Core Efficacy

**DOI:** 10.3390/molecules28207132

**Published:** 2023-10-17

**Authors:** Baixin Kou, Yuxin Jiang, Yanan Chen, Jingrong Yang, Jin Sun, Yan Yan, Lili Weng, Chunping Xiao

**Affiliations:** College of Pharmacy, Changchun University of Chinese Medicine, Changchun 130117, China; koubaixin@163.com (B.K.); jjyyxx2023@163.com (Y.J.); 18043919903@163.com (Y.C.); 13894298316@163.com (J.Y.); sunjin0509@163.com (J.S.); 13843212250@163.com (Y.Y.)

**Keywords:** Chinese herbal medicines, Gentianae Radix et Rhizoma, class, LC–MS/MS, bioefficacy, acute liver injury, Nrf2/HO-1

## Abstract

(1) Background: Establishment of a method for evaluating Gentianae Radix et Rhizoma (GRR) classes based on chemical composition and core efficacy; (2) Methods: Liquid chromatography–mass spectrometry (LC–MS) was used to determine the chemical constituents of GRR-first class (GF) and GRR-second class (GS). The cell viability, liver function, oxidative stress enzyme activity, and inflammatory factor levels of GF and GS on H_2_O_2_-induced HepG2 cells were determined with CCK-8, ELISA, and biochemical methods, and the antioxidant activity of the two was evaluated using bioefficacy; ELISA, biochemical methods, real-time fluorescence quantitative polymerase chain reaction (RT-qPCR) method, and Western blot (WB) were used to determine the liver function, oxidative stress enzyme activity, inflammatory factor levels, and expression of related genes and proteins in mice with acute liver injury (ALI) model induced with 0.3% CCl_4_ olive oil solution after gavage administration; (3) Results: GF and GS had the same types of components, but the cyclic enol ether terpenes such as morinlon goside c, loganin, gentiopicroside, and swertiamarin differed significantly between the two; the effect of GF on CCl_4_-induced acute hepatic injury in C57BL/6 mice was stronger compared to GS. It helped alleviate weight loss, increase hepatic and splenic indices, improve hepatic lobular structure and hepatocyte status, inhibit collagen deposition, enhance oxidative stress and anti-inflammatory-related genes and protein expression, and decrease apoptotic genes and proteins more significantly than GS; (4) Conclusions: In this study, we established a GRR class evaluation method combining chemical composition and core medicinal effects, which can rapidly determine the differential composition of GF and GS, detect the quality of GRR through antioxidant bioefficacy, and validate it with in vivo experiments, which provides references for the evaluation of the class of GRR and the rational use of medication in the clinic.

## 1. Introduction

Chinese herbal medicines (CHMs) are affected by a variety of factors such as environment, harvesting, processing, and clinical use, which often leads to uncontrollable quality of CHMs, which is one of the challenges faced in the clinical practice of CHMs [1]. In order to solve this dilemma, it is necessary to establish scientific and up-to-date quality standards for CHMs. At present, China’s current commodity specification and grade standard for Chinese herbal medicines is the “Commodity Specification Standard for 76 Types of CHM”, which was promulgated in 1984, and the test contents mainly include traits, smell, color, and so on. At the present stage, methods for evaluating the quality of CHMs include fingerprinting [2], near-infrared spectroscopy [3], genome sequencing [4], etc., and the main objects of analysis are the differences in appearance traits [5] (e.g., color, odor, diameter, weight, etc.), microscopic structure [6], single chemical components [7], secondary metabolites [8], etc. However, there is a relative gap in the research on combining the multi-indicator components and core efficacy of CHMs as the basis for grade classification, which cannot be adapted to the development of the research on CHMs at this stage. In order to improve the mechanism of class research, it is crucial to establish an evaluation system for the class of CHMs, that combines quantitative and effective effects.

Gentianae Radix et Rhizoma (GRR) is a perennial herb of the genus *Gentian* in the family Gentianaceae, which was first recorded in the Shennong’s Classic of the Materia Medica in the Han Dynasty (206 B.C.–220 A.D.), and its efficacy is to protect the liver and benefit the gallbladder, which has been used for thousands of years [9,10]. In accordance with the above method of quality evaluation of traditional Chinese medicine (TCM), the group preliminarily graded the GRR according to their quantity, diameter size, color, check items, water-soluble leachate, and gentiopicroside content [11]. Since the ultimate goal of research on CHMs is to ensure the rationality of clinical application, the core efficacy should be taken into account when classifying classes and constructing quality standards, but there are few relevant reports at this stage. In order to meet the requirements of modernization and internationalization of TCM, it is imperative to supplement and improve the quality standards of TCM with new technologies. Studies have shown [12] that GRR contains a variety of components such as iridoids, flavonoids, polysaccharides, etc., and the iridoids are the key pharmacological material basis. Therefore, it is imminent to find out the compositional differences between GRR-first class (GF) and GRR-second class (GS). In addition, the core efficacy of GRR is hepatoprotective, and in vivo studies include the alleviation of hepatic injury caused by substances such as CCl_4_ [13], acetaminophen [14], lipopolysaccharide [15], and α-naphthyl isothiocyanate [16] or lipid accumulation induced by acute and chronic alcohol intake [17]. Whereas, in vitro studies include the protective effect of GRR against ouabain and H_2_O_2_-induced HepG2 cell damage [18,19]. H_2_O_2_-induced HepG2 cells and CCl_4_-induced C57BL/6 mice were in vitro and in vivo models of acute liver injury (ALI), respectively, which coincidentally corresponded to the properties of GRR’s action on the liver; so, the above models were chosen to evaluate the core efficacy of GRR. 

In summary, the current research on the class of GRR is only limited to the chemical composition, which is not related to the biological activity. In order to ensure the quality and efficacy of GRR, it is indispensable to establish a comprehensive evaluation system integrating composition, efficacy, and mechanism. The aim of this study is to improve the evaluation method of GRR and other CHMs, to provide a basis for the selection, quality control, and rational use of CHMs, to ensure the quality and efficacy of CHMs, and to provide an idea to promote the modernization and internationalization of CHMs.

## 2. Results

### 2.1. Differences in Composition between GF and GS

#### 2.1.1. Differences in Traits between GF and GS

As clearly shown in Figure 1, the surface of GF is dark gray-brown. The GF displays the following traits: long roots, diameter ≥ 0.20 cm, the number of roots ≥ 15, yellowish white or yellowish surface, yellowish white on part of the skin section, uniform length and thickness, complete, no broken joints, moth, and mold. Whilst, the GS displays the following traits: dark gray-brown or dark brown surface, light yellow or yellow-brown root surface, yellow-white or light yellow-brown on part of the skin section, length and thickness are not uniform, incomplete, residual broken joints are present, no insect, and mildew.

#### 2.1.2. Compound Analysis and Identification

Results from the negative ion mode revealed that a total of 53 chemical components were identified from GF and GS. GF and GS had the same types of components, but the contents were different. Among them, loganin, isovitexin, 8-Epiloganic acid, eustomoside, swertiamarin, gentiopicroside, and 46 other components were found to be more abundant in GF than GS, and GF was lower than GS in 7-*O*-glucose-isoorientin, trifloroside and seven other ingredients (Figure 2A,B). Combined with the control product (Figure 2C), the average contents of gentiopicroside, swertiamarin, and sweroside were calculated to be 5.29, 0.45, and 0.77 mg/mL, respectively, for GF; and the average contents of gentiopicroside, swertiamarin, and sweroside were calculated to be 2.84, 0.37, and 0.26 mg/mL, respectively, for GS. Detailed information is shown in Table 1.

#### 2.1.3. Principal Component Analysis (PCA) of GF and GS

The raw data obtained for GF and GS were imported into Marker View™ 1.2.1, and the corresponding parameters were set to remove the isotope ion peaks. The processed data were imported into SIMCA 14.1 for unsupervised PCA, and the scatter plots of the scores are shown in Figure 2D. The GF and GS were each clustered in quadrants one and four and two and three, and there was no crossover between the two groups, indicating that there were differences in the compositions of the GF and GS.

#### 2.1.4. Orthogonal Partial Least Squares-Discriminant Analysis (OPLS-DA) of GF and GS

The model was further tested using the OPLS-DA test, and it was found that GF and GS were distributed on both sides of the *Y*-axis, with obvious zoning, indicating that there were significant differences in the chemical compositions of GF and GS (Figure 2E). The model was subjected to 200 replacement tests, and R^2^ = 0.992 and Q^2^ = 0.626 in the negative ion mode, indicating that the results were significant and reliable. Two regression lines had larger slopes, and the R^2^ and Q^2^ obtained from the left randomized arrangement were smaller than the original values on the right, which indicated that the model was stable and reliable, with strong predictive ability and no overfitting, as shown in Figure 2F.

Further analysis of the above data showed that the components with VIP values > 1 were peak **41** (morinlongoside c), peak **8** (isovitexin), peak **7** (1,3,7-Trihydroxy-4,8-dimethoxystigmasterone), peak **49** (4‴-*O*-β-d-Glucopyranosylscabraside), peak **21** (eustomoside), peak **23** (swertiamarin), peak **35** (kogen glycolide), peak **10** (8-Epiloganic acid), peak **33** (gentrigeoside a), peak **15** [2′-(2,3-Dihydroxybenzoyl)-gentianoside], peak **24** (6-Keto-8-acetyl-Leptoside), peak **19** (secoxyloganin), peak **2** (loganin), peak **22** (6-*O*-d-glu gentiopicroside), peak **47** (6‴-*O*-β-d-glucopyranosyltrifloroside), peak **26** (gentiopicroside), peak **48** (2′-o-caffeoylloganin), peak **39** (strychnic acid 11-*O*-β-glucopyranosyl ester), peak **37** [4″-*O*-β-d-glucopyranosyl-6′-*O*-(4-*O*-β-d glucopyranosylcaffeoyl)linearoside], peak **20** (morroniside), peak **16** (glu-caffeic acid), peak **18** (1-*O*-β-d-glucopyranosyl-4-epiamplexine), peak **32** (6′-*O*-glucose gentiopicroside), peak **12** (trilobatin), and peak **17** (eustomorusside).

In summary, morinlongoside c, loganin, gentiopicroside, and swertiamarin differed significantly between GF and GS, and can be used as differentiating components between the two (Figure 2G).

### 2.2. Effect of GF and GS on Hydrogen Peroxide-Induced Oxidative Stress Model in HepG2 Cells

#### 2.2.1. Effect of GF and GS on HepG2 Cells under the Effect of Hydrogen Peroxide

The effect of GF and GS on the viability of HepG2 cells under the action of hydrogen peroxide was determined, and the results are shown in Figure 3B, which shows that the cell viability of the M group was significantly decreased (*p* < 0.01) compared with the N group, indicating that H_2_O_2_ had a significant damaging effect on HepG2 cells. Compared with the M group, cell viability was significantly higher in the administered group (*p* < 0.01). Cell viability was significantly reduced in the following: low-, medium-, and high-dose groups of GS compared to the high-dose group of GF (*p* < 0.01); low- and medium- dose-groups of GF (*p* < 0.01); and control cell viability group, but not significantly different (Figure 3A).

#### 2.2.2. Effects of GF and GS on Liver Function Indexes, Inflammatory Factors and Oxidative Stress Indexes of HepG2 Cells under the Action of H_2_O_2_

Compared with the N group, the ALT and AST levels were significantly increased in the M group (*p* < 0.01), indicating that the stimulation of H_2_O_2_ led to the abnormal liver function of HepG2 cells. Compared with the model group, ALT and AST levels were significantly down-regulated in the FH, FM, FL, SH, and SM groups (*p* < 0.01), and ALT levels were significantly reduced in the SL group (*p* < 0.01), but no decreasing trend was observed in AST levels. Compared with the FH group, ALT and AST levels were significantly higher in the SM and SL groups (*p* < 0.01) (Figure 3C,D). The levels of IL-1 and IL-6 were significantly higher in the M group compared with the N group (*p* < 0.01). The levels of IL-1 and IL-6 were significantly lower in all dosing groups compared to the M group. The levels of IL-1 and IL-6 were significantly higher (*p* < 0.01) in the FM, FL, SH, SM, and SL groups compared with the FH group (Figure 3E,F). The levels of SOD and CAT were significantly lower in the M group compared with the N group (*p* < 0.01). Compared with the M group, SOD levels showed a significant increase in all dosing groups (*p* < 0.01), and CAT levels were also significantly higher in all groups except for the SL group (*p* < 0.01). Finally, SOD and CAT values were significantly lower in all other GRR administration groups compared to the FH group (Figure 3G,H).

#### 2.2.3. Results of Bioavailability Testing of GF and GS

Based on the above method, a test method for the antioxidant bioefficacy of GRR was constructed, with which the differences in antioxidant activity of six batches of GRR were detected, and the results of the relative proliferation rates are shown in Figure 4A, which shows that there were large differences in the relative proliferation rates of HepG2 cells in response to the H_2_O_2_-induced oxidative stress in the six batches of different classes of GRR. The results were entered into the “TCM Potency Calculator” to calculate the bioefficacy of different classes of GRR, and the results are shown in Figure 4B. The data showed that the antioxidant activity potency of different classes of GRR was in the range of 381.692–907.193 U·mg^−1^, with a difference of nearly two-fold between the highest and the lowest potency, and the extreme difference was 525.501 U·mg^−1^. The average value of antioxidant activity was 658.754 U·mg^−1^, indicating that the biological quality of different classes of GRR varied greatly.

### 2.3. Protective Effects of GF and GS on Mice with ALI

#### 2.3.1. Effects of GF and GS on Body Weight of Mice with ALI

As shown in Figure 5A,B, no significant difference in body weight was observed in each group at 1 d, which ranged from 18–22 g. At 15 d, the body weight of the model M group was extremely significantly reduced compared with that of the N group (*p* < 0.01). Compared with the M group, the body weights of the C, FH, FL, and SH groups were all extremely significantly higher (*p* < 0.01); while the body weight of the SL group was significantly higher (*p* < 0.05). In addition, on the 15th d, compared with the FH group, the body weights of the FL and SL groups were both extremely significantly lower (*p* < 0.01), and the body weight of the SH group was significantly lower (*p* < 0.05).

#### 2.3.2. Effects of Gentian GF and GS on Liver and Spleen Indexes in Mice with ALI

The results are shown in Figure 5C,D. Compared with the N group, there was a significant difference in the elevated liver and spleen indices in the M group of mice (*p* < 0.01). Compared with the M group, the liver and spleen indices were highly significantly reduced in the C, FH, FL, and SL groups (*p* < 0.01). Liver and spleen indices were significantly increased in the SH group (*p* < 0.05) and highly significantly increased in the SL group (*p* < 0.01) when compared to the FH group.

#### 2.3.3. Histopathologic Examination of the Liver

HE staining results showed that the M group of mice had intact hepatic lobule structure, neatly arranged hepatocytes, clear nucleus structure, and no swelling and inflammatory infiltration of cells. The hepatocytes of mice with acute liver injury were significantly degenerated, with swollen hepatocytes, transparent cytoplasm, necrosis of most cells, and inflammatory cell infiltration. Compared with the M group, the C group and all doses of GRR administration group could significantly reduce the degree of liver injury, as demonstrated by the neatly arranged hepatocytes (Figure 5E) and the more similar structure of the liver lobules and the state of the hepatocytes of mice in the FH group to those of the M group.

Sirius scarlet staining showed that a large number of collagen fibers proliferated in the confluent area of the liver tissue of CCl_4_-model mice, forming fibrous intervals, and the structure of hepatic lobules was damaged and pseudolobules were formed. Collagen deposition was only seen in the confluent area in the C group and different classes of GRR-dose groups, and the degree of collagen deposition was significantly reduced, and the improvement of collagen deposition was more obvious with the intervention of the FH group (Figure 5F).

#### 2.3.4. Effects of GF and GS on SOD, CAT, ALT, AST, IL-1, and IL-6 Levels in Mice with ALI

Compared with the N group, SOD and CAT levels were significantly and extremely reduced in the M group of mice (*p* < 0.01). Compared with the M group, SOD and CAT levels were extremely significantly higher (*p* < 0.01) in the C, FH, and SH groups; and significantly higher (*p* < 0.05) in the FL and SL groups, as shown in Figure 6A,B. In addition, compared with the FH group, the SH group had a highly significant decrease in SOD level (*p* < 0.01) and significantly lower CAT values (*p* < 0.05), which indicated that FH was more effective in terms of antioxidant effects (Figure 6A,B).

The results of liver function indexes in mice are shown in Figure 6C,D. Compared with the N group, the levels of ALT and AST in mice in the M group were highly significantly elevated (*p* < 0.01). It indicated that the mice in the M group showed liver function impairment. Compared with the M group, ALT and AST were significantly lower in the C, FH, FL, SH, and SL groups of mice (*p* < 0.05). In addition, compared with the FH group, the ALT and AST levels in the SH group were significantly higher (*p* < 0.05 for ALT level and *p* < 0.01 for AST level).

Compared with the N group, the hepatic tissue levels of IL-1 and IL-6 were both highly significantly and extremely elevated in the M group of mice (*p* < 0.01). Compared with the M group, IL-1 and IL-6 levels were extremely significantly lower (*p* < 0.01) in the C, FH, SH, and FL groups; whereas, IL-1 and IL-6 levels were significantly lower (*p* < 0.05) in the SL group, as shown in Figure 6E,F. In addition, compared with the FH group, the SH group showed a significant increase in IL-6 and IL-1 levels (*p* < 0.05), which indicated that GF was superior to GS in terms of anti-inflammatory effects.

#### 2.3.5. Effects of GF and GS on the Expression Levels of Nrf2, HO-1, TLR4, FAS, ERK, and FXR mRNA in Mice with ALI

As shown in Figure 7A,B,E, Nrf2, HO-1, and FXR mRNA expression levels were significantly lower in the M group compared to the N group (*p* < 0.01). Compared with the M group, the Nrf2, HO-1, and FXR mRNA expression levels were significantly higher in each dosing group (*p* < 0.01 or *p* < 0.05). Compared with the FH group, the Nrf2, HO-1, and FXR mRNA expression levels were significantly lower in the SL group (*p* < 0.01). Compared with the N group, Fas, TLR4, and ERK mRNA expression levels were significantly higher in the M group (*p* < 0.01). Compared with the M group, the Fas mRNA expression level was significantly lower in all dosing groups (*p* < 0.01), except for the SL group, whereby its TLR4 and ERK mRNA expression levels were significantly lower (*p* < 0.01). Fas, TLR4, and ERK mRNA expression levels were significantly higher (*p* < 0.01) in the SL group compared with the FH group (Figure 7C,D,F).

#### 2.3.6. Effects of GF and GS on Protein Expression of HO-1, Nrf2, IκBα and TLR4 in Mice with ALI

As shown in Figure 8, the expression of HO-1 and Nrf2 proteins in the M group was highly and significantly reduced compared with the N group (*p* < 0.01), indicating that carbon tetrachloride affected the antioxidant function of C57BL/6 mice. Compared with the M group, the expression of HO-1 and Nrf2 proteins in the FH, FL, and SH groups was significantly higher (*p* < 0.01), indicating that GRR administration significantly increased the expression of oxidative stress proteins. Compared with the FH group, the expression of HO-1 and Nrf2 proteins was significantly lower in the FL and SL groups (*p* < 0.01), while for the SH group, the expression of HO-1 was significantly lower (*p* < 0.05), and the expression of Nrf2 was highly and significantly lower (*p* < 0.01). This suggests that the effect of GF is stronger in enhancing the antioxidant capacity.

As shown in Figure 9, compared with the N group, IκBα expression was highly and significantly lower (*p* < 0.01) and TLR4 expression was highly and significantly higher (*p* < 0.01) in the M group, indicating that the modeling drug CCl_4_ causes inflammatory injury in mice, which results in the low expression of nuclear factor inhibitory proteins and the high expression of Toll-like receptors on the surface of the cell membrane. Compared with the M group, the expression of IκBα was significantly higher in each administration group (*p* < 0.05), and the expression of TLR4 was significantly lower in all administration groups (*p* < 0.01), except for the SL group. IκBα expression was significantly lower (*p* < 0.01) and TLR4 expression was significantly higher (*p* < 0.01) in the other administered groups compared to the FH group. This suggests that GF is more effective in enhancing anti-inflammatory capacity.

## 3. Discussion

The evaluation of the grade of Chinese herbal medicines is mainly based on the identification of traits, microscopic, physicochemical, and biometric identification [39], and with the deepening of the research, the traditional identification methods alone can not be adapted to the development of the times. And because of the complexity and diversity of the components of TCM, the study of the differences in chemical composition is essential in the process of class evaluation. Although some people have carried out studies on the differences in the composition of *Gentiana scabra* and *Gentiana rigescens* [24], there are few reports on the studies on the class of GRR and the differences in the composition of GRR multi-indicators. In order to evaluate the quality of different grades of GRR, this study established a method for evaluating the class of GRR based on chemical constituents and core efficacy. The results showed that GF and GS components were of the same type, and morinlongoside c, loganin, gentiopicroside, and swertiamarin differed significantly between the two, which can be used as indicator components for evaluating the quality of GRR. Some studies have shown that morinlongoside c, as a good antioxidant, is widely found in CHMs [40,41]. Loganin administration showed a protective effect against apoptosis in the liver of type 2 diabetic mice [42]. Gentiopicroside prevents alcoholic liver injury [43]. Swertiamarin can attenuate liver injury in nonalcoholic fatty liver disease mice through the pathway of up-regulation of the Nrf2/HO-1 pathway and down-regulation of reactive oxygen species binding protein-1 [44]. The above further illustrates that GRR achieves hepatoprotective effects through multiple pathways, including antioxidants, inhibition of apoptosis, and reduction of inflammation. In addition, GF was stronger than GS in protecting liver function, inhibiting inflammation, and antioxidant bioefficacy in HepG2 cells under the action of H_2_O_2_. Moreover, the inhibitory effects of GF on body weight loss, reduction of hepatic and splenic indices, abnormalities of hepatic lobular structure and hepatocyte status, and collagen deposition in CCl_4_-induced acutely hepatic-injured C57BL/6 mice were significantly better than those of GS, which was similar to Zhang’s study [45]. Therefore, it was hypothesized that due to the significant enrichment of the above components in GF, the efficacy of GF was superior to that of GS.

The Nrf2/HO-1 signaling pathway is involved in oxidative stress and is associated with a variety of diseases such as ischemia-reperfusion and apoptosis [46]. FXR is a ligand-mediated transcription factor that is highly expressed in the liver and plays a regulatory role in oxidative stress and inflammatory responses [47]. The Fas pathway is one of the major regulatory pathways of apoptosis, which has an important impact on the balance between cell proliferation and apoptosis. ERK and TLR4 signaling pathways can activate related proteins, leading to the activation of inflammatory factors such as IL-1 and IL-6, with the end result of liver injury [48,49]. RTq-PCR results showed that compared with mice under the effect of GF, the expression levels of Nrf2, HO-1, and FXR in GS were significantly reduced (*p* < 0.01), and the expression levels of Fas, TLR4, and ERK were significantly increased (*p* < 0.01). This indicates that the ability of GF to increase the expression of genes related to oxidative stress and decrease the expression of genes related to apoptosis and lipid accumulation was superior to that of GS, which is consistent with the previous results of hepatoprotective effects. In addition, the WB results showed that GF was able to decrease TLR4 protein expression by increasing Nrf2-, HO-1- and IκBα-related protein expression, which again confirmed that the efficacy of GF was superior to that of GS, further indicating that gentian was able to resist hepatic injury from multiple pathways such as oxidative stress, inflammation, and apoptosis, and that the effect of GF was superior to that of GS.

## 4. Materials and Methods

### 4.1. Reagents

Dulbecco’s Modified Eagle Medium medium (DMEM, Gibco, Billings, MT, USA, batch no.: 8121342), Catalase kit (CAT, batch no.: 20221008), Superoxide Dismutase kit (SOD, batch no.: 20221008) were purchased from Beijing Solepol Science and Technology Co., Ltd., Beijing, China, Cell Counting Kit-8 (CCK-8, Beijing Gold Clone, Beijing, China, batch no.: 20210531), vitamin C tablets (Harbin Pharmaceutical Group Sanjingmingshui Pharmaceutical Co., Ltd., Harbin, China, lot number: State Pharmaceutical License H20067527, specification: 50 mg), RNA Extraction Kit (lot number: AL40413A), Reverse Transcription Kit (lot number: AL50948A), and cDNA Amplification Kit (lot number: AL61810A) were purchased from TaKaRa; CCl_4_ solution (Shanghai Macklin Biochemical Technology Co. Ltd., Shanghai, China, Lot No. C14485128), olive oil (No. 69, Nonglin Road, Futian District, Shenzhen, China, Lot No. 223283089), interleukin-1 (IL-1), interleukin-6 (IL-6), alanine aminotransferase (ALT), aspartate transaminase (AST) kits were purchased from Jiangsu Enzyme Immunity Industry Ltd., Yancheng, China (batch no.: 202302); Nuclear Factor erythroid 2-Related Factor 2 (Nrf2, batch no.: 16396-1-AP), heme oxygenase 1 (HO-1. Batch No. 10701-1-A2), Inhibitory Subunit of NF-κBα (IKBα, Batch No. 00095768), Recombinant Toll-Like Receptor 4 (TLR4, Batch No. 110521211223), Beta-actin (lot number: 20536-AP) and secondary antibody (lot number: SA00001-2) were purchased from Proteintech, Rosemont, IL, USA.

### 4.2. Chinese Medicinal Herbs

GRR is the dried root and rhizome of *Gentiana scabra* Bunge., and three batches each of GF and GS were obtained according to the pre-grading method of the subject group.

### 4.3. Sample Preparation

#### 4.3.1. Preparation of Aqueous Extract of GRR

The aqueous extract of GRR was prepared according to the method of the previous research group: 3.0 kg of GRR herb was weighed, and decocted with water two times (the mass ratio of water and herb was 10:1 and 8:1, respectively), each time for 1.5 h. The filtrate was combined two times, and the solution was concentrated into 1.2 g/mL (in terms of the amount of raw drug), which was reserved for the cellular and pharmacological experiments.

#### 4.3.2. Preparation of Test Solution

The aqueous extract of gentian under Section 4.3.1 was used as raw material, which was distilled and concentrated under reduced pressure at 60 °C to obtain the extract (the yield was about 50%. A total of 0.1 g extract (equivalent to 0.2 g of raw drug) was dissolved with 10 mL of 50% methanol to prepare a 0.02 g/mL (raw drug amount) solution. After that, the above solution was passed through a 0.22 μm filter membrane, and the liquid phase vial was left to be measured (GF and GS were prepared according to the above method).

#### 4.3.3. Preparation of Control Solution

Gentiopicroside, dangonin, and swertiamarin were precisely weighed and added with an appropriate amount of 50% methanol to dissolve, respectively, to make a mixed standard solution containing 1 mg/mL each of gentiopicroside, dangonin, and swertiamarin.

### 4.4. Determination of Chemical Composition of Different Classes of GRR by LC-MS/MS

#### 4.4.1. Chromatographic Conditions

Reference was made to Zhang’s research method [24] and changes were made as follows: chromatographic column: Agilent ZORBAX SB-C18 (4.6 × 250 mm, 5 μm); mobile phase: 0.1% formic acid in water (A)–acetonitrile (B) with a gradient of 0–10 min, 10–20% B; 10–25 min, 20–53% B; 25–30 min, 53–80% B; 30–35 min, 80–100% B; and 35–42 min, 100% B; column temperature: 30 °C; flow rate: 1.0 mL/min; and injection volume: 10 μL.

#### 4.4.2. Mass Spectrometry Conditions

In negative ion mode, the ion spray voltage was 2.7 kV, the capillary temperature was 320 °C, the source heater temperature was 200 °C, the sheath gas (N_2_) was fifteen arbitrary units, the auxiliary gas (N_2_) was eight arbitrary units, and the purge gas (N_2_) was two arbitrary units. The Orbitrap analyzer performed a full sweep of the MS mass number ranging from m/z 50 to 1345 with a resolution of 30,000 (FWHM defined as *m*/*z* 400). The MS data were recorded in contour plot format.

#### 4.4.3. Compound Identification Method

The database of gentian-related components was established by consulting Science Direct, Pubmed, Medline, and other databases, including compound names, relative molecular masses, molecular formulae, chemical structures, and other information [43,50,51,52,53,54,55]; and the compounds were structurally resolved by combining the fragmentation information of the secondary mass spectrometry, retention time, related literature and databases.

### 4.5. Cellular Experiments

#### 4.5.1. CCK-8 Method to Detect the Protective Effect of GF and GS on Cells

When the cell number was at 5 × 10^4^/mL, the cells were seeded in 96-well plates with a final volume of 200 μL per well and cultured at 37 °C and 5% CO_2_ for 24 h. The medium was discarded, and the cells were divided into nine groups: the normal group (DMEM), the model group (DMEM + 1450 μM H_2_O_2_), the control group (DMEM + 1450 μM H_2_O_2_ + 80 mg/mL vitamin C tablets), GF high-dose group (FH, DMEM + 1450 μM H_2_O_2_ + 4 mg/mL GF), GF medium-dose group (FM, DMEM + 1450 μM H_2_O_2_ + 2 mg/mL GF), GF low-dose group (FL, DMEM + 1450 μM H_2_O_2_ + 1 mg/mL GF), and GS high-dose group (SH, DMEM + 1450 μM H_2_O_2_ + 4 mg/mL GS), GS medium-dose group (SM, DMEM + 1450 μM H_2_O_2_ + 2 mg/mL GS), and GS low-dose group (SL, DMEM + 1450 μM H_2_O_2_ + 1 mg/mL GS). According to the above grouping, the above cells were cultured at 37 °C and 5% CO_2_ for 0.5 h. PBS was added and rinsed once. A total of 10 μL CCK-8 solution was added to each well to avoid air bubbles, and the plate was incubated at 37 °C and 5% CO_2_ for 40 min, and then the culture supernatant was carefully aspirated from the wells. Subsequently, the OD value of each well was measured using a full-wavelength multifunctional enzyme labeling instrument at the wavelength of 450 nm and the cell survival rate was calculated. The experiment was repeated at least three times.

#### 4.5.2. Crystalline Violet Staining

Cells were seeded in 6-well plates at 1 × 10^5^·L^−1^, 2 mL per well, and evenly distributed. After 24 h, each well was treated according to Section 4.5.1, and after 0.5 h, the medium was aspirated and rinsed with PBS three times, Then, 500 μL of prepared crystal violet staining solution was added to each well, and then incubated for 15 min in a 37 °C incubator. After the incubation, pictures were taken under a fluorescence inverted microscope.

#### 4.5.3. Effects of GF and GS on Liver Function Indexes, Inflammatory Factors, and Oxidative Stress Indexes of HepG2 Cells under the Action of Hydrogen Peroxide

Cells with good growth conditions were taken and seeded in 6-well plates at 2.5 × 10^5^/mL, 2 mL per well, and nine groups under Section 4.5.1 were set up simultaneously. After 0.5 h, the cells were washed twice with 1 mL/well of PBS, then 1 mL/well of trypsin was added to digest the cells in each well, and 1 mL/well of complete medium was added to the cells after complete digestion. After which, the cells were collected in 5 mL EP tubes and centrifuged at 800 r/min for 5 min. For the determination of SOD and CAT, the cells were broken in 1.5 mL EP tubes (placed on ice), and for the determination of IL-1, IL-6, ALT, and AST, the supernatants were centrifuged in 1.5 mL EP tubes (placed on ice). The activity of liver function markers, inflammatory factors, and oxidative stress markers in the cells were measured using the kit according to the manufacturer’s instructions.

#### 4.5.4. Determination of Antioxidant Potency of GF and GS

The aqueous extract of gentian obtained under Section 4.3.1 was used as the test material (T), and vitamin C tablets were used as the reference material, the processing method is as follows: A total of 0.4 g Vc reagent was weighed precisely, and then dissolved by shaking in 5 mL distilled water to obtain the vitamin C reference material (S) with a final concentration of 80 mg/mL. The prepared test and reference solutions were diluted three times according to the 2:1 agent spacing. HepG2 cells were cultured from Petri dishes for passaging, and then dispensed into 96-well plates to continue culture. At a cell density of 75%, the T/S group, respectively, was added with a medium containing the test/reference at a volume ratio of 10 μL/200 μL to pretreat HepG2 cells for 0.5 h. The control group (C) was incubated with a normal medium. The medium was discarded, and each group was replaced with a medium containing H_2_O_2_ 1450 μM and incubated for 0.5 h. Absorbance was measured at 450 nm using the CCK-8 method. The relative proliferation rate = (AT or S/AC − 1) × 100% was measured three times in parallel. The relative proliferation rate was used as a parameter to calculate the antioxidant potency of the test article according to the “Traditional Chinese Medicine Potency Calculator (developed and researched by the former 32nd Institute of TCM of the People’s Liberation Army)”, with the potency of the reference substance as 100 U·mg^−1^.

### 4.6. Animal Management

#### 4.6.1. Modeling Drug Administration and Material Collection

Seventy C57BL/6 mice (all SPF grade, weighing 20–22 g, purchased from Liaoning Changsheng Biotechnology Co., Ltd., Shenyang, China, QC No. SCXK2020-0001) were divided into a normal group (N, n = 10, indiscriminate), a model group (M, n = 10, CCl_4_), a positive control group (C, n = 10, CCl_4_ + bifendatatum), a GF high-dose group (FH, n = 10, CCl_4_ + 0.78 g/Kg GF), GF low-dose group (FL, n = 10, CCl_4_ + 0.39 g/kg GF), GS high-dose group (SH, n = 10, CCl_4_ + 0.78 g/Kg GS), and GS low-dose group (SL, n = 10, CCl_4_ + 0.39 g/kg GS). All mice were first acclimatized for 7 days, and the N and M groups were administered saline by gavage, whereas the C group was administered an aqueous solution of bifendatatum at a concentration of 150 mg/kg by gavage at 10 mL/kg body weight (BW). In this study, mice in each GRR administration group were administered by gavage at a dose of (high) 0.78 g/kg, or 0.39 g/kg (low) (10 mL/kg) once a day for 14 consecutive days in all groups, and the BW of the mice was recorded daily during the experiment. After 2 h of gavage administration on the last day, all groups except the N group were injected with 10 mL/kg BW of 0.3% CCL_4_ olive oil solution to establish a liver injury model, while the N group was injected with olive oil solution without CCl_4_ as a control [56]. The mice were then fasted but provided with water supply for 16 h and then weighed. After anesthesia, blood was collected from mice for serum preparation. Livers and spleens were dissected after decapitation, rinsed with ice-cold saline, and weighed. Half of the liver tissue was routinely fixed and embedded in a wax block, and the other half was placed in liquid nitrogen for 1 min and then stored at −80 °C. This study was approved by the Ethics Committee of Changchun University of Chinese Medicine under the ethical review approval number 2023005.

#### 4.6.2. Liver Histopathology

While the mice were executed, a portion of liver tissue was quickly collected and fixed in 4% paraformaldehyde. Next, a small piece of tissue was excised, embedded, dehydrated, made into a paraffin block, sectioned (3–5 μm thick), stained with hematoxylin and eosin (HE) and stained with aspergillus scarlet, and observed histopathologically with a light microscope.

#### 4.6.3. Serum Biochemistry to Determine SOD, CAT, AST, ALT, IL-1 and IL-6 Levels

Prior to the execution of the mice, peripheral blood was collected by removing the eyeballs in sterile tubes, supernatants were taken and then stored in a refrigerator at −80 °C. Serum SOD and CAT levels were measured using different biochemical kits according to the manufacturer’s instructions. Chopped liver tissue (10 mg) was mixed with 900 μL of phosphate-buffered saline (PBS) (containing protease inhibitors) and homogenized thoroughly on ice. To further lyse the tissue, the homogenate was broken using cryosonication. Finally, the homogenate was centrifuged at 5000 rpm for 5–10 min and the supernatant was collected for cytokine measurements. Inflammatory factor levels such as AST, ALT, IL-1, and IL-6 in liver tissue were measured using ELISA kits according to the manufacturer’s instructions.

#### 4.6.4. Detection of HO-1, Nrf2, TLR4, Fas, ERK, and FXR mRNA Expression Levels in Mouse Liver Tissues

Real-time fluorescence quantitative polymerase chain reaction (RT-qPCR) was used for detection. HO-1, Nrf2, TLR4, Fas (Fatty acid synthetase), ERK (extracellular regulated protein kinases), and FXR (Farnesoid × Receptor) mRNA primers were purchased from TAKRA (Table 2). The total RNA was extracted from the liver tissues of mice in each group by liquid nitrogen grinding, then reverse transcribed into cDNA according to the method of reverse transcription kit instruction, and then amplified by PCR. The PCR reaction conditions were pre-denatured at 95 °C for 30 s, denatured at 95 °C for 3 s, and finally annealed at 60 °C for 30 s, with a total of 40 cycles. The expression level of the target gene was calculated by the 2^−ΔΔCt^ method using β-actin as an internal reference.

#### 4.6.5. Detection of HO-1, Nrf2, IκBα, and TLR4 Protein Expression in Mouse Liver Tissue

The assay was performed by the Western blot (WB) method. The total protein was extracted from mouse liver tissue by adding an appropriate amount of lysate and placed on ice for 30 min, centrifuged at 1200 r/min for 15 min at 4 °C, and the supernatant was taken to determine the protein concentration by bicinchoninic acid (BCA) method. The protein was denatured by boiling, then subjected to sodium dodecyl sulfate-polyacrylamide gel electrophoresis, and the membrane was transferred; the membrane was covered with 5% skimmed milk powder for 1.5 h, and then HO-1, Nrf2, IκBα, TLR4, and β-actin primary antibodies were added (all at a dilution of 1:1000), and the membrane was washed with Tris-HCl buffer salt + Tween-20 (TBST) three times, each time for 10 min, and then the membrane was incubated with a secondary antibody (at a dilution of 1:2000) at room temperature for 1.5 h. The membrane was washed three times, each time for 10 min, and the secondary antibody (at a dilution of 1:2000) was added and incubated for 1 h at room temperature. The membrane was washed three times with TBST for 10 min each time, and the secondary antibody (dilution of 1:2000) was added and incubated at room temperature for 1 h. The membrane was washed three times with PBS containing 1% Tween-20 for 10 min each time, and Electrochemiluminescence (ECL) reagent was added, and the membrane was imaged with a fully automated chemiluminescence imaging analyzer. Image J 6.0 software was used for the analysis, and the expression level of the target protein was expressed as the ratio of the gray value of the target protein to that of the internal reference β-actin.

### 4.7. Software

LC–MS solution software, Simca 14.1 software (https://www.prueapp.cn/html/1000198, accessed on 14 October 2023), Graph Prism 9.0 (https://graphpad-prism.cn/?c=p&a=shopList, accessed on 14 October 2023), Image J software (https://imagej.net/imagej-wiki-static/Welcome, accessed on 14 October 2023), and Mview slice visualization software (https://www.freewarepocketpc.net/ppc-download-mview-v1-35.html, accessed on 14 October 2023) were used.

### 4.8. Data Analysis

SPSS 25.0 software was used to statistically analyze the data, and the results were expressed as mean ± standard deviation. One-way ANOVA was used for comparison between multiple groups, and the LSD-t test was used for a two-by-two comparison between groups. The test level was α = 0.05. β-actin was used in the ΔCt method as an internal reference.

## 5. Conclusions

In this study, we established a gentian grade evaluation method combining chemical composition and core efficacy, and were able to rapidly determine the differential composition of GF and GS, detect the quality of GRR through antioxidant bioefficacy, and validate the core efficacy of the two in in vivo experiments in terms of epigenetic indexes, liver function, and molecular mechanisms, and similar conclusions were obtained. Therefore, it is recommended to choose GF for clinical use.

## Figures and Tables

**Figure 1 molecules-28-07132-f001:**
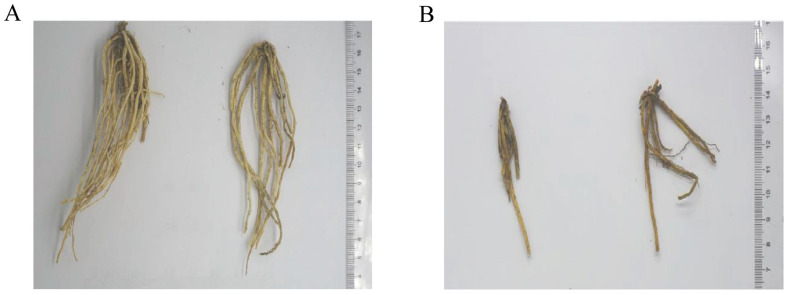
Differences in traits between GF and GS ((**A**) GF trait map; (**B**) GS trait map).

**Figure 2 molecules-28-07132-f002:**
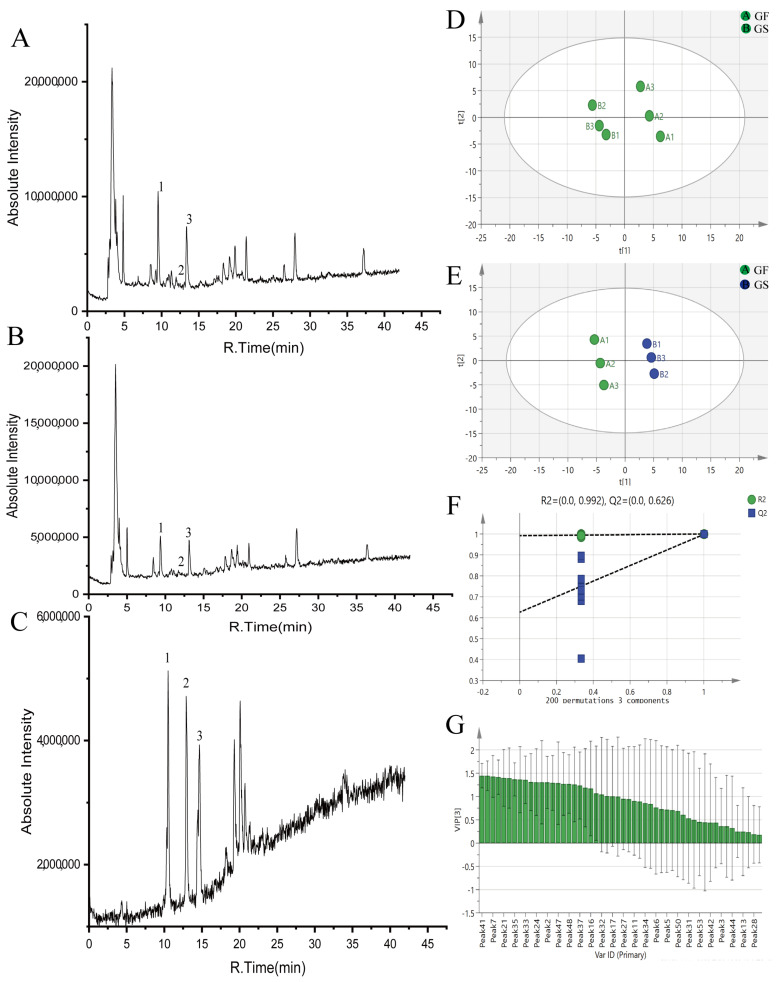
Compound analysis and identification of GF and GS ((**A**) total ion flow diagram of GF; (**B**) total ion flow diagram of GS; (**C**) standard product diagram; (**D**) PCA scores of GF and GS; (**E**) OPLS-DA scores of GF and GS; (**F**) 200 substitutions of the OPLS-DA models of GF and GS; and (**G**) VIP plots for GF and GS). Note: **1** is gentiopicroside, **2** is dangonin, and **3** is swertiamarin.

**Figure 3 molecules-28-07132-f003:**
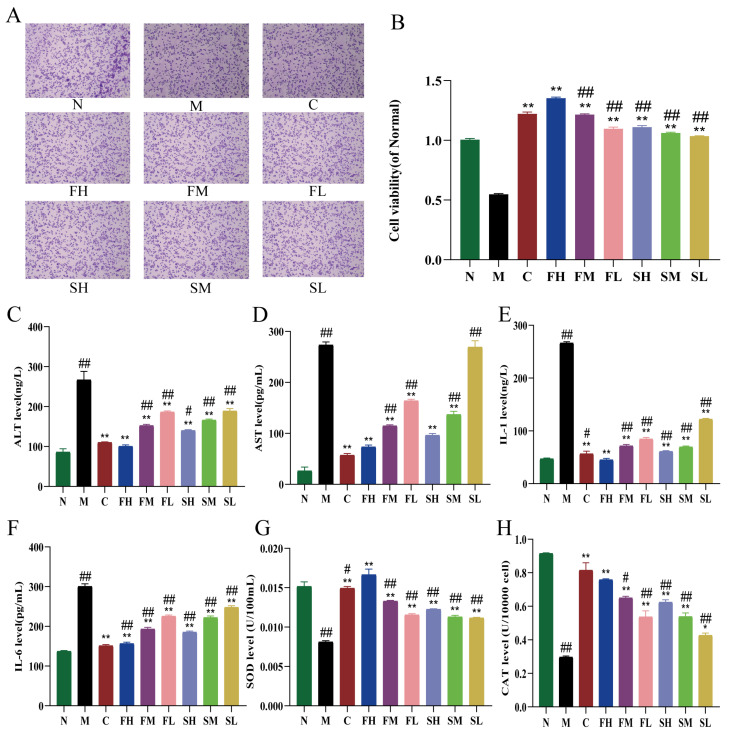
Effects of GF and GS on HepG2 cells under the action of hydrogen peroxide ((**A**) effects of GRR on the morphology of oxidative stress cells in HepG2 cells induced by H_2_O_2_, ×400; (**B**) effects of GF and GS on the viability of oxidative stress cells in HepG2 cells induced by H_2_O_2_; (**C**,**D**) the levels of ALT and AST, respectively; (**E**,**F**) the levels of IL-1 and IL-6, respectively; and (**G**,**H**) the levels of SOD and CAT, respectively). Note: N (normal group), M (model group), C (control group), FH (GF high-dose group), FM (GF middle-dose group), FL (GF low-dose group), SH (GS high-dose group), SM (GS middle-dose group), and SL (GS low-dose group). ** *p* < 0.01, * *p* < 0.05 compared with the model M group; ^##^
*p* < 0.01, ^#^
*p* < 0.05 compared with the FH group.

**Figure 4 molecules-28-07132-f004:**
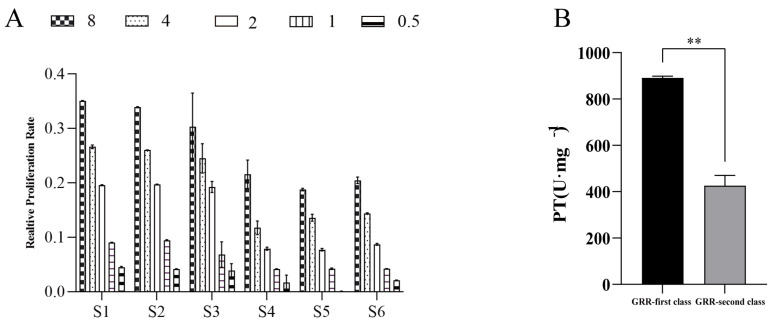
GF and GS bioefficacy assay results ((**A**) relative proliferation rate results; (**B**) GF and GS bioefficacy results). Note: S1–S3 are GF, S4–S6 are GS; and 0.5, 1, 2, 4, and 8 represent the concentrations of GF and GS. ** *p* < 0.01 represents a significant difference compared with GF.

**Figure 5 molecules-28-07132-f005:**
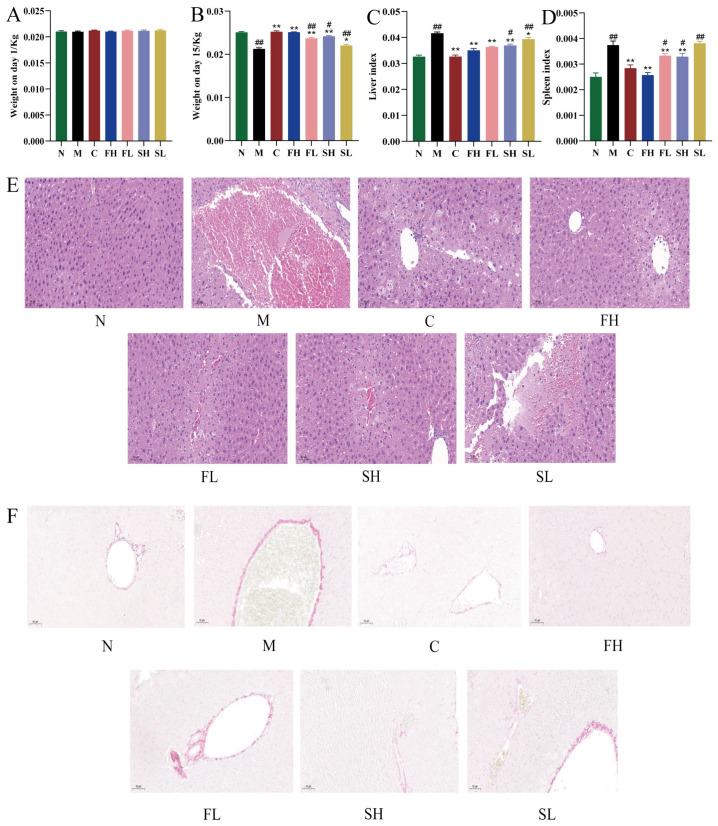
Protective effects of GF and GS on mice with ALI ((**A**) effects of GF and GS on body weight of mice with acute liver injury at 1 d; (**B**) effects of GF and GS on body weight of mice with acute liver injury at 15 d; (**C**) effects of different classes of GRR on hepatic indices of mice with acute liver injury; (**D**) effects of different classes of GRR on splenic indices of mice with acute liver injury; (**E**) effects of GF and GS on HE staining results of liver tissues of mice with acute liver injury, ×200; and (**F**) effects of GF and GS on the results of Sirius scarlet staining of liver tissues of mice with acute liver injury, ×200). Note: ** *p* < 0.01, * *p* < 0.05 compared with the model M group; ^##^
*p* < 0.01, ^#^
*p* < 0.05 compared with the FH group. N (normal group, n = 10, indiscriminate), M (model group, n = 10, CCl_4_), C (positive control group, n = 10, CCl_4_ + bifendatatum), FH (GF high dose-group, n = 10, CCl_4_ + 0.78 g/Kg GF), FL (GF low-dose group, n = 10, CCl_4_ + 0.39 g/kg GF), SH (GS high-dose group, n = 10, CCl_4_ + 0.78 g/Kg GS), and SL (GS low-dose group, n = 10, CCl_4_ + 0.39 g/kg GS).

**Figure 6 molecules-28-07132-f006:**
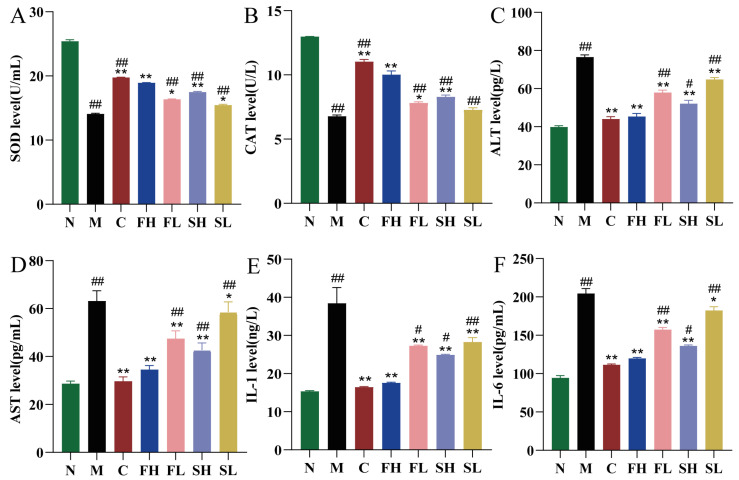
Effects of GF and GS on SOD, CAT, ALT, AST, IL-1, and IL-6 levels in mice with ALI ((**A**) SOD; (**B**) CAT; (**C**) ALT; (**D**) AST; (**E**) IL-1; and (**F**) IL-6). Note: ** *p* < 0.01, * *p* < 0.05 compared with the M group; ^##^
*p* < 0.01, ^#^
*p* < 0.05 compared with the FH group. N (normal group, n = 10, indiscriminate), M (model group, n = 10, CCl_4_), C (positive control group, n = 10, CCl_4_ + bifendatatum), FH (GF high-dose group, n = 10, CCl_4_ + 0.78 g/Kg GF), FL (GF low-dose group, n = 10, CCl_4_ + 0.39 g/kg GF), SH (GS high-dose group, n = 10, CCl_4_ + 0.78 g/Kg GS), and SL (GS low-dose group, n = 10, CCl_4_ + 0.39 g/kg GS).

**Figure 7 molecules-28-07132-f007:**
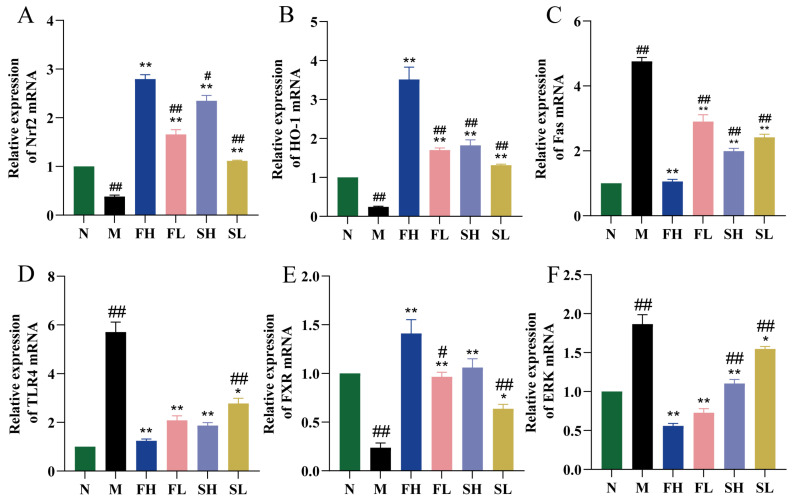
Effects of GF and GS on the expression levels of Nrf2, HO-1, TLR4, Fas, ERK, and FXR mRNA in mice with ALI. ((**A**) Nrf2; (**B**) HO-1; (**C**) TLR4; (**D**) Fas; (**E**) ERK; and (**F**) FXR). Note: ** *p* < 0.01, * *p* < 0.05 compared with the model M group; ^##^
*p* < 0.01, ^#^
*p* < 0.05 compared with the FH group. N (normal group, n = 10, indiscriminate), M (model group, n = 10, CCl_4_), FH (GF high-dose group, n = 10, CCl_4_ + 0.78 g/Kg GF), FL (GF low-dose group, n = 10, CCl_4_ + 0.39 g/kg GF), SH (GS high-dose group, n = 10, CCl_4_ + 0.78 g/Kg GS), and SL (GS low-dose group, n = 10, CCl_4_ + 0.39 g/kg GS).

**Figure 8 molecules-28-07132-f008:**
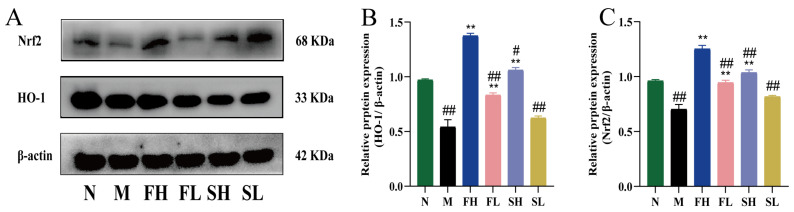
Effects of GF and GS on HO-1 and Nrf2 protein expression in mice with ALI. ((**A**) HO-1 and Nrf2 gel imaging image; (**B**) Relative expression of HO-1; (**C**) Relative expression of Nrf2). Note: ** *p* < 0.01, compared with the model M group; ^##^
*p* < 0.01, ^#^
*p* < 0.05 compared with the FH group. N (normal group, n = 10, indiscriminate), M (model group, n = 10, CCl_4_), FH (GF high-dose group, n = 10, CCl_4_ + 0.78 g/Kg GF), FL (GF low-dose group, n = 10, CCl_4_ + 0.39 g/kg GF), SH (GS high-dose group, n = 10, CCl_4_ + 0.78 g/Kg GS), and SL (GS low-dose group, n = 10, CCl_4_ + 0.39 g/kg GS).

**Figure 9 molecules-28-07132-f009:**
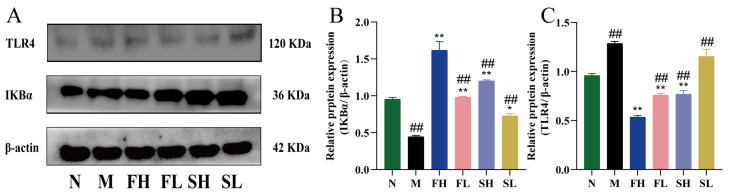
Effects of GF and GS on IκBα and TLR4 protein expression in mice with ALI. ((**A**) IκBα and TLR4 gel imaging image; (**B**) Relative expression of IκBα; (**C**) Relative expression of TLR4). Note: ** *p* < 0.01, * *p* < 0.05 compared with the model M group; ^##^
*p* < 0.01, compared with the FH group. N (normal group, n = 10, indiscriminate), M (model group, n = 10, CCl_4_), FH (GF high-dose group, n = 10, CCl_4_ + 0.78 g/Kg GF), FL (GF low-dose group, n = 10, CCl_4_ + 0.39 g/kg GF), SH (GS high-dose group, n = 10, CCl_4_ + 0.78 g/Kg GS), and SL (GS low-dose group, n = 10, CCl_4_ + 0.39 g/kg GS).

**Table 1 molecules-28-07132-t001:** Analyzing and identifying the common chemical constituents of GF and GS.

No.	t	Name	Molecular Formula	Ionization Mode	*m*/*z*	Fragmentation Ion	A1 ^1^	A2 ^1^	A1/A2	Literature
**1**	0.27	ethyl gallate	C_9_H_10_O_5_	[M + FA − H]	242.900	197/98	27,202	24,505	1.11	[20]
**2**	2.90	loganin	C_11_H_16_O_5_	[M + FA − H]	273.050	227/113	266,412	105,047	2.54	[21]
**3**	3.34	3-episwertiajaposide c	C_17_H_24_O_10_	[M − H]	387.250	193	1,178,076	1,043,432	1.13	[22]
**4**	3.37	2′-*O*-(p-Coumaroyl) loganin	C_27_H_34_O_12_	[M − H]	549.350	513/256	3,874,778	2,847,413	1.36	[23]
**5**	3.96	2′-*O*-(2,3-dihyben)-swertamairn	C_23_H_26_O_13_	[M − H]	509.300	254	562,969	312,224	1.80	[24]
**6**	6.45	10-Hydro-9-hydroxysweroside	C_16_H_24_O_10_	[M + FA − H]	421.400	375/187	111,956	49,269	2.27	[25]
**7**	7.04	1,3,7-Trihydroxy-4,8-dimethoxystigmasterone	C_15_H_12_O_7_	[M + FA − H]	349.200	303/151	39,312	13,895	2.83	[26]
**8**	7.19	isovitexin	C_21_H_20_O_10_	[M + FA − H]	477.150	431/215	46,038	13,171	3.50	[27]
**9**	7.67	tianmu dihuangoside a	C_15_H_22_O_8_	[M + FA − H]	375.300	327/163	35,038	25,641	1.37	/
**10**	7.77	8-Epiloganic acid	C_16_H_24_O_10_	[M − H]	375.200	187	49,097	16,812	2.92	[28]
**11**	7.91	secologanoside	C_16_H_22_O_11_	[M − H]	389.250	194	40,310	20,481	1.97	[24]
**12**	8.01	trilobatin	C_21_H_24_O_10_	[M − H]	435.300	217	30,176	18,759	1.61	[29]
**13**	8.14	vanilloloside	C_13_H_16_O_9_	[M − H]	315.250	157	334,352	409,370	0.82	[24]
**14**	8.18	mangiferin	C_19_H_18_O_11_	[M − H]	421.250	212/210	1,144,870	437,852	2.61	[24]
**15**	8.69	2′-(2,3-dihydroxybenzoyl)-gentiopicroside	C_23_H_24_O_12_	[M + FA − H]	537.350	491/245	159,081	69,087	2.30	[30]
**16**	8.99	glu-caffeic acid	C_15_H_18_O_9_	[M − H]	341.350	170	59,945	39,519	1.52	[24]
**17**	9.07	eustomorusside	C_16_H_24_O_12_	[M − H]	407.400	205	80,748	57,239	1.41	[31]
**18**	10.04	1-*O*-β-d-Glucopyranosyl-4-epiamplexine	C_16_H_26_O_9_	[M + FA − H]	407.300	203	195,450	69,172	2.83	/
**19**	10.43	secoxyloganin	C_19_H_16_O_10_	[M + FA − H]	449.300	405/201	252,412	97,601	2.59	[24]
**20**	10.61	morroniside	C_17_H_26_O_11_	[M − H]	451.350	407/204	275,463	141,987	1.94	[24]
**21**	10.78	eustomoside	C_16_H_22_O_11_	[M − H]	389.250	389/194	500,658	164,340	3.05	[31]
**22**	11.01	6-*O*-d-glu gentiopicroside	C_22_H_30_O_14_	[M − H]	563.400	517/258	601,199	189,514	3.17	[24]
**23**	11.63	swertiamarin	C_16_H_22_O_10_	[M + FA − H]	419.300	373/186	924,346	307,549	3.01	[24]
**24**	11.93	6-keto-8-acetyl-leptoside	C_17_H_24_O_11_	[M + FA − H]	449.350	403/201	235,341	112,837	2.09	/
**25**	12.14	6‴-*O*-β-d-glucopyranosyltrifloroside	C_22_H_30_O_14_	[M + FA − H]	563.250	517/258	51,399	39,543	1.30	[32]
**26**	12.27	gentiopicroside	C_16_H_20_O_9_	[M + FA − H]	401.200	355/177	2,998,944	1,797,106	1.67	[24]
**27**	12.47	6′-*O*-vanilloyl-8-epikingiside	C_25_H_30_O_14_	[M − H]	553.500	276	50,572	30,357	1.67	/
**28**	12.71	3′-*O*-acetylsweroside	C_18_H_24_O_10_	[M − H]	399.200	199	64,698	63,471	1.02	[33]
**29**	13.97	7-*O*-glucose-isoorientin	C_27_H_26_O_17_	[M + FA − H]	639.400	621/310	23,688	52,604	0.45	/
**30**	14.23	dangonin	C_16_H_22_O_9_	[M + FA − H]	403.350	357/178	92,849	61,409	1.51	[24]
**31**	14.53	syringin	C_17_H_24_O_9_	[M − H]	371.250	185	46,683	33,030	1.41	[24]
**32**	15.28	6′-*O*-glu gentiopicroside	C_22_H_30_O_14_	[M − H]	497.600	258	3850	43,159	0.89	[24]
**33**	15.38	gentrigeoside a	C_36_H_60_O_12_	[M − H]	683.450	341	288,876	83,816	3.45	[24]
**34**	15.62	caryptoside	C_17_H_26_O_11_	[M − H]	405.450	404/202	57,483	25,322	2.27	/
**35**	16.23	kogen glycol	C_20_H_50_O_2_	[M + FA − H]	487.450	321/160	82,442	32,002	2.58	[24]
**36**	16.41	gentianidine	C_9_H_6_O_4_	[M + FA − H]	223.200	177/88	52,216	20,032	2.61	/
**37**	16.53	4″-*O*-β-d-Glucopyranosyl-6′-*O*-(4-*O*-β-d glucopyranosylcaffeoyl) linearoside	C_46_H_56_O_25_	[M − H]	1007.600	503	278,267	70,719	3.93	/
**38**	17.18	erythricine	C_10_H_9_NO_2_	[M + FA − H]	220.100	174/86	50,212	47,581	1.06	/
**39**	17.24	strychnic acid 11-*O*-β-glucopyranosyl ester	C_22_H_34_O_15_	[M − H]	539.400	537/268	140,303	77,311	1.81	[34]
**40**	17.38	amaropanin	C_29_H_30_O_12_	[M − H]	569.300	284	58,034	56,590	1.03	/
**41**	17.61	morinlongoside c	C_22_H_32_O_15_	[M + FA − H]	602.550	535/267	39,725	14,593	2.72	/
**42**	18.60	trifloroside	C_35_H_42_O_20_	[M + FA − H]	826.800	781/390	223,134	265,386	0.84	/
**43**	19.21	acremoxanthone d	C_36_H_60_O_10_	[M + FA − H]	697.350	651/325	70,327	41,369	1.70	/
**44**	20.75	6,7-dehydro-8-acetyl-rhamnoside	C_17_H_24_O_10_	[M + FA − H]	433.300	387/193	4230	36,541	1.16	/
**45**	21.30	rehmannioside c	C_30_H_42_O_17_	[M + FA − H]	740.650	673/336	122,115	99,402	1.23	[35]
**46**	22.79	dedihydroxybenzoate-macrophylloside	C_33_H_40_O_19_	[M − H]	739.100	369	138,156	142,501	0.97	/
**47**	24.18	6‴-*O*-β-d-Glucopyranosyltrifloroside	C_41_H_52_O_25_	[M − H]	943.550	471	37,499	24,574	1.53	[36]
**48**	26.24	2′-*O*-Caffeoylloganin	C_27_H_34_O_13_	[M + FA − H]	610.750	565/282	146,350	75,618	1.94	/
**49**	28.95	4‴-*O*-β-d-Glucopyranosylscabraside	C_46_H_54_O_25_	[M − H]	1005.550	502	267,168	112,441	2.38	[36]
**50**	30.65	rindoside	C_35_H_42_O_21_	[M + FA − H]	843.400	797/398	70,154	49,064	1.43	[37]
**51**	32.54	pinetoxanthone	C_25_H_24_N_6_O	[M + FA − H]	424.500	412/205	111,474	73,624	1.51	[38]
**52**	32.64	tianmu dihuangoside e	C_15_H_22_O_8_	[M − H]	329.400	164	347,026	598,268	0.58	/
**53**	36.30	deglu-trifloroside	C_29_H_32_O_15_	[M − H]	619.350	309	45,047	49,987	0.90	[24]

^1^ A1 is the average peak area of GF and A2 is the average peak area of second-class GS.

**Table 2 molecules-28-07132-t002:** Primer sequences for qPCR.

Gene	Forward Primer Sequence (5′ → 3′)	Reverse Primer Sequence (5′ → 3′)
β-actin	CATTGCTGACAGGATGCAGAAG	TGCTGGAAGGTGGACAGTGAGG
HO-1	CAGAGTTTCTTCGCCAGAGG	TGAGTGTGAGGACCCATCG
Nrf2	ATCCTTTGGAGGCAAGACAT	TCCTGTTCCTTCTGGAGTTG
TLR4	TAAGTGCCGAGTCTGAGTGTAA	AACCCTTATTGTCATTCCCAG
Fas	ATGTCCGGGATCTGGGTTCACTTGT	TTAAACCAAGTTTTCACTTTCATT
ERK	ACCGTGACCTCAAGCCTTCC	GATGCAGCCCACAGACCAAA
FXR	AGGGGTGTAAAGGTTTCTTCAGGA	ACACTTTCTTCGCATGTACATATCCAT

Note: β-actin is the reference gene; HO-1 is the heme oxygenase 1 gene; Nrf2 is the Nuclear Factor erythroid 2-Related Factor 2 gene; TLR4 is the recombinant Toll-Like Receptor 4 gene; Fas is the Fatty acid synthetase gene; ERK is extracellular regulated protein kinases gene; and FXR is Farnesoid X Receptor gene.

## Data Availability

Data are contained within the article.

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
