# Peer review of "A Study of Gentianae Radix et Rhizoma Class Differences Based on Chemical Composition and Core Efficacy"

_molecules, 2023, doi:10.3390/molecules28207132_

Round 1
Reviewer 1 Report
The article is interesting, well-integrated and well-discussed, with a clear novelty. The following recommendations are suggested.
Abstract. OK
Introduction. OK
Materials & Methods. Is GGR sun-dried by farmers? Please include the LC-MS/MS system specifics in section 4.4. The specification of the identification process included in the result section must be relocated to the methodology section. Please explain the equivalence between the doses used in the in vitro and the in vivo models, as well as the the projected equivalence for humans. Include the description of the genes in the footnote of Table 2.
Results. Why authors did not quantified compounds by the external standard method using standards of those components commercially available or by normalizing data with internal standard? Please clarify the level of identification in Table 1 and lines 95-99, Page 3. Why authors only marked three identified component in Figure 2? Please justify a multivariate analysis with three replicates, and further clarify if those are biological replicates in the footnote. Include the legend description in the footnote of Figure 4. A scale must be included in the microphotographs of the histology analysis. The distribution of the graphs included in Figure 6 do not allow a clear data visualization.
Discussion. OK
Conclusion. OK
Additional notes. The authors declared that they obtained the informed consent from all the subjects, what are they referring to?
Author Response
Dear Editor and Reviewer,
On behalf of my co-authors, we thank you very much for giving us an opportunity to revise our manuscript, and we also appreciate reviewers very much for their positive and constructive comments and suggestions on our manuscript entitled “A study of Gentianae Radix et Rhizoma class differences based on chemical composition and core efficacy” (Manuscript Number: 2670743).
We revised the manuscript according to these comments and suggestions. In general, we have tried our best to revise our manuscript and provide the point-by-point responses. All changes were marked by highlighting using the “Track Changes” function in the revised manuscript. Attached please find our responses to the referees’ comments.
The following is a summary list of changes:
- The English language in the revised manuscript has been carefully corrected to improve grammar and readability. Errors in detail about units and the problem of missing spaces have been corrected.
- 2. Point-by-point response to Comments and Suggestions for Authors
Comments 1: Is GRR sun-dried by farmers?
Response 1: Thank you for pointing this out. I agree with this comment. Therefore, GRR was purchased from China Jilin Province Beiyao Processing Company, which is a professional Chinese herbal medicine processing company.
Comments 2: Please include the LC-MS/MS system specifics in section 4.4. The specification of the identification process included in the result section must be relocated to the methodology section.
Response 2: Agree. We have, accordingly, revised “2.1.2” section and we have integrated the content of Section 2.1.2 into section 4.4.3. : Compound identification method.
Comments 3: Please explain the equivalence between the doses used in the in vitro and the in vivo models, as well as the the projected equivalence for humans.
Response 3: Agree. We have, accordingly, revised as following:
(1) Dose basis of in vitro model : According to the cell concentration screening experiment ( concentration range of 0.125,0.25,0.5,1,2,4,8,16,32 mg / mL ), the results showed that there was no significant effect on the survival rate of HepG2 cells below 8 mg / mL. Considering the economic and pharmacodynamic factors, 1 mg / mL, 2 mg / mL and 4 mg / mL were selected as the low, medium and high dose groups.
(2) The dose basis of in vivo model : the conversion of human and mice was 0.0026. The Chinese Pharmacopoeia stipulates that the dosage of gentian is 3-6 g (crude drug). Therefore, 6 g is high dose (clinical dosage is low dose, 2 times is high dose ). The high dose is converted to the dose of mice : 6 × 0.0026 / 0.02 = 0.78 g / Kg, low dose : 0.78 / 2 × 2 = 0.39 g / Kg (crude drug). Therefore, the equivalent dose of gentian to humans is 3-6 g per day.
Comments 4: Include the description of the genes in the footnote of Table 2.
Response 4: Thank you for pointing this out. I agree with this comment. Therefore, We have detailed annotations under Table 2.
Comments 5: Why authors did not quantified compounds by the external standard method using standards of those components commercially available or by normalizing data with internal standard?
Response 5: Thank you for pointing this out. I agree with this comment. Because the standard we used is the commercially available and qualified standard, this study uses the external standard method to calculate the peak area of GF and GS, and the concentration and peak area of the standard are known to obtain the concentration of GF and GS.
Comments 6: Please clarify the level of identification in Table 1 and lines 95-99, Page 3.
Response 6: Thank you for pointing this out. I agree with this comment. Based on a large number of databases and literature surveys, we identified and identified the basic information of the compounds by Shimadzu LC-MS analysis software.
Comments 7: Why authors only marked three identified component in Figure 2?
Response 7: Thank you for pointing this out. I agree with this comment. Because the three components ' gentiopicroside, dangonin, and swertiamarin ' in Figure 2 are the most widely reported in the literature and the most abundant, common and classic components in Gentiana scabra.
Comments 8: Please justify a multivariate analysis with three replicates, and further clarify if those are biological replicates in the footnote. Include the legend description in the footnote of Figure 4.
Response 8: Thank you for pointing this out. I agree with this comment. 0.5 and 1, 2, 4 and 8 mg/mL represent the concentrations of GF and GS administered, which are biological replicates in the footnotes (n = 3). And we have modified the corresponding position in the article.
Comments 9: A scale must be included in the microphotographs of the histology analysis.
Response 9: Thank you for pointing this out. I agree with this comment. We have re-uploaded histological analysis microphotographs with scales.
Comments 10: The distribution of the graphs included in Figure 6 do not allow a clear data visualization.
Response 10: Thank you for pointing this out. I agree with this comment. We have corrected the graphical distribution of Figure 6.
Comments 11: The authors declared that they obtained the informed consent from all the subjects, what are they referring to?
Response 11: Thank you for pointing this out. I agree with this comment. We chose this one wrongly. All experiments should be ethically reviewed. This article takes mice as the research object.

Reviewer 2 Report
The manuscript compares the efficacy of two classes of Gentianae Radix et Rhizoma (GRR). GRR is an herb that has been used for thousands of years in traditional Chinese medicine and has shown hepatoprotective effects. In order to obtain the best possible treatment results, it is necessary to determine the quality standard of the raw herbal material. Therefore, a comparison of two classes of GRR was made, determining their chemical composition, antioxidant activity and basic efficacy as to epigenetic indicators, liver function and molecular mechanisms. Thus, a methodology that can be used to evaluate other herbal medicines is presented.
The manuscript is interesting and presents an important issue, but needs some remarks.
Minor remarks
Throughout the manuscript, the abbreviations GF and GS or FG and SG are used alternately. You should decide on one type of abbreviation and use it consistently throughout the text.
The terms in vitro and in vivo should be italicized.
Table 1 needs to be corrected to make it more readable. The "Name" column should be narrower, while the "m/z," "A1," "A2," "A1/A2" columns should be - wider.
Figure 2 - I do not understand why Figure 2B is rotated 180° relative to Figures A and C, there is no justification for this. This should be corrected.
Figures 2D, 2E - there is no information on what the letters A and B stand for. This should be supplemented.
In line 156 is the phrase "Cell viability was significantly and significantly...". The word "significantly" is repeated twice, I believe one of them was meant to sound different. Please correct this.
The sentence in lines 203-207 should be rephrased, as it is now barely understandable.
Items 2.3.1 and 2.3.2 have the same title, I think they should be different.
Author Response
Another Point-by-point response to Comments and Suggestions for Authors
Comments 1: Throughout the manuscript, the abbreviations GF and GS or FG and SG are used alternately. You should decide on one type of abbreviation and use it consistently throughout the text.
Response 1: Thank you for pointing this out. I agree with this comment. Therefore, We have corrected Table 1 to make it more readable.
Comments 2: The terms in vitro and in vivo should be italicized.
Response 2: Agree. We have, accordingly, revised Figure 2B to make it normal.
Comments 3: Table 1 needs to be corrected to make it more readable. The "Name" column should be narrower, while the "m/z," "A1," "A2," "A1/A2" columns should be - wider.
Response 3: Agree. We have, accordingly, revised Table 1.
Comments 4: Figure 2 - I do not understand why Figure 2B is rotated 180° relative to Figures A and C, there is no justification for this. This should be corrected.
Response 4: Thank you for pointing this out. I agree with this comment. Therefore, We have revised Figure 2B.
Comments 5: Figures 2D, 2E - there is no information on what the letters A and B stand for. This should be supplemented.
Response 5: Thank you for pointing this out. I agree with this comment. We have marked A and B of Fig.2D and E, which are GF and GS, respectively.
Comments 6: In line 156 is the phrase "Cell viability was significantly and significantly...". The word "significantly" is repeated twice, I believe one of them was meant to sound different. Please correct this.
Response 6: Thank you for pointing this out. I agree with this comment. We have corrected ' Cell viability was significantly and significantly... ' to ' Cell viability was significantly... '.
Comments 7: The sentence in lines 203-207 should be rephrased, as it is now barely understandable.
Response 7: Thank you for pointing this out. I agree with this comment. We have deleted the unclear statements of lines 203-207.
Comments 8: Items 2.3.1 and 2.3.2 have the same title, I think they should be different.
Response 8: Thank you for pointing this out. I agree with this comment. The title at 2.3.2 has been corrected to ' Effects of gentian GF and GS on liver and spleen indexes in mice with ALI '.
Once again, thank you very much for your comments and suggestions. And we hope that the revised manuscript can be accepted by Molecules. If further revision is necessary, please contact me at: koubaixin@163.com. In addition, we did not find the 4.10 part you said, but you refer to the 4.1 part. We have reduced the weight of this part. If there is a need for further modification, please contact me.
Thank you and best regards.

Round 2
Reviewer 1 Report
Clearly state that compounds were putatively identified in table 4, since authors assigned a compound name as if all components identification was validated by comparison with commercial standard.
Author Response
Dear Editor and Reviewer,
On behalf of my co-authors, we thank you very much for giving us an opportunity to revise our manuscript, and we also appreciate reviewers very much for their positive and constructive comments and suggestions on our manuscript entitled “A study of Gentianae Radix et Rhizoma class differences based on chemical composition and core efficacy” (Manuscript Number: 2670743).
We revised the manuscript according to these comments and suggestions. In general, we have tried our best to revise our manuscript and provide the point-by-point responses. All changes were marked by highlighting using the “Track Changes” function in the revised manuscript. Attached please find our responses to the referees’ comments.
The following is a summary list of changes:
- Point-by-point response to Comments and Suggestions for Authors
Comments 1: Clearly state that compounds were putatively identified in table 4, since authors assigned a compound name as if all components identification was validated by comparison with commercial standard.
Response 1:Thank you for pointing this out. I agree with this comment. Therefore, we have checked the name, m/z and source of the compounds in Table 1, supplemented the references and corrected the ambiguous names. The revised part is shown in Table 1, highlighted.
Once again, thank you very much for your comments and suggestions. And we hope that the revised manuscript can be accepted by Molecules. If further revision is necessary, please contact me at: koubaixin@163.com.
Thank you and best regards.
Sincerely yours,
Lili Weng email: weng1969@sohu.com (L.W.);
Chunping Xiao email: btxnw@163.com (C.X.)
